# A Uniformity Index for Precipitation Particle Axis Ratios Derived from Radar Polarimetric Parameters for the Identification and Analysis of Raindrop Areas

Yue Sun [1] , Hui Xiao [1,2,*], Huiling Yang [1], Haonan Chen [3] , Liang Feng [1] , Weixi Shu [1,2] and Han Yao [4]

1 Key Laboratory of Cloud-Precipitation Physics and Severe Storms (LACS), Institute of Atmospheric Physics, Chinese Academy of Sciences, Beijing 100029, China
2 College of Earth Sciences, University of Chinese Academy of Sciences (UCAS), Beijing 100049, China
3 Department of Electrical and Computer, Colorado State University, Fort Collins, CO 80523, USA
4 Qingdao Meteorology Bureau, Qingdao 266003, China
* Correspondence: hxiao@mail.iap.ac.cn; Tel.: +86-10-82995318

**Abstract:** A uniformity index for the axis ratios ($U_{ar}$) derived from dual polarization weather radar data is proposed for raindrop area identification and analysis. The derivation of this new parameter is based on radar scattering simulations and assumptions. $U_{ar}$ is between 0 and 1 and can be calculated from the differential reflectivity ($Z_{DR}$) and the copolar correlation coefficient ($\rho_{hv}$), which reflects the uniformity of the axis ratio ($r$) of the particle group. For raindrops, $U_{ar}$ is close to 1 under ideal conditions, but is clearly different from that of ice particles whose value is close to 0. Studies conducted during two convective weather events observed by X-band and S-band radar are presented to show the $U_{ar}$ features. In convective areas, high $U_{ar}$ presents a U-shaped vertical structure. One branch corresponds to the $Z_{DR}$ column, while the other branch is located at the rear of the convective cloud zone and is lower in altitude, representing the process of ice particles melting into raindrops and then being transported upward by a strong updraft. In stratiform cloud areas, a more than 95% overall identification ratio is obtained when the threshold of $U_{ar}$ is set to 0.2~0.3 for discriminating rain layers.

**Keywords:** dual polarization weather radar; axis ratio; rain area identification

## 1. Introduction

The distribution and variation characteristics of the phase state (liquid, ice, mixed phase, etc.) of hydrometeors in clouds are extremely important issues in precipitation physics. Dual polarization weather radar obtains polarimetric variables, such as the horizontal/vertical reflectivity factor ($Z_H/Z_V$), differential reflectivity ($Z_{DR}$), copolar correlation coefficient ($\rho_{hv}$) and differential propagation phase shift ($K_{DP}$), which are closely related to the microphysical properties of hydrometeors in clouds [1–3]. For example, large raindrops show a flat shape and a corresponding high $Z_{DR}$ value when under air resistance [4]. This is clearly different from the various shapes of tumbling hail and graupel, which make it possible to roughly distinguish liquid and solid particles in the high $Z_H$ region [5]. Furthermore, lower $Z_H$ appears in snow and ice crystals due to the lower dielectric constant, while lower $\rho_{hv}$ appears in mixed phase particles and sometimes in ice phase particles due to the variation of dielectric constant, shape and orientation [3,6]. Since different kinds of particles are not easily and directly identified due to the overlapping of the range of polarization parameters, a hydrometeor classification algorithm (HCA) based on the fuzzy logic algorithm [6] is the most feasible solution to obtain a general qualitative result. This kind of algorithm has been developed and improved over the past 20 years [7–16]. However, the HCA still has limitations in terms of subjectivity and experience [16]. Thus, its results cannot be regarded as absolutely accurate or the only microphysical analysis

results, nor can the HCA completely replace the analysis of the original observed variables and other analysis methods. In particular, an additional input temperature profile is needed in most of the methods above, which means that the original polarimetric observables still cannot completely identify the hydrometeor phase independently.

Another principle algorithm that can distinguish the phase state of hydrometeor particles in clouds involves identifying the melting layer (ML) so that the part below the ML is identified as the rain area. In stratiform mixed phase clouds, the ML exhibits a $Z_H$ peak in the vertical direction [17,18], known as the bright band (BB) in radar meteorology. There is also a $Z_{DR}$ peak and a valley of $\rho_{hv}$ in the ML. These dual polarization weather radar signals are closely related to changes in the dielectric constant, particle density, size and shape during the falling of ice phase particles [3]. Weather radar mostly adopts the polar coordinate volume scanning mode, where automatic ML detection algorithms are mainly built based on radar image features, including the gradient and extreme of $Z_H$ in a vertical profile of reflectivity (VPR) [18–20], the boundary of high $\rho_{hv}$ [21], the thresholds or gradient of $Z_H$, $Z_{DR}$ and $\rho_{hv}$ [22–25], and the matching degree with the ideal model [26]. Automatic ML detection results can reduce contamination in radar quantitative precipitation estimation (QPE). Furthermore, ML is helpful for studying cloud and precipitation physical processes such as the ML sinking due to the riming or coalescence of snow [3,27,28]. However, such ML features mainly exist in large-scale stable stratiform precipitation and are difficult to be identified in convective clouds with severe temporal and spatial variability. Hence, the current algorithms above are not easily applied to the study of the melting or freezing processes within convective clouds. In addition, the accuracy of these algorithms mostly also depends on the additional input temperature profile, and it is usually necessary to summarize the thresholds of multiple variables. Therefore, it is still meaningful to find a more accurate and reliable method or some variables based on weather radar data to identify the hydrometeor phase.

In this study, a new parameter involving the microphysical characteristics of hydrometeor particles is proposed. The new parameter is derived from existing polarimetric radar observables and is found to reflect the uniformity of precipitation particles' axial ratio. By this parameter alone, a simpler method for identifying raindrop areas is presented by setting a threshold. The derivation and demonstration of the new parameter will aim at the S-band (wavelength 10 cm) and X-band (wavelength 3.2 cm), which are commonly used in weather radars. The S-band is the most common band of operational weather radar, which has little rain attenuation and a long detection distance. X-band radar often has a smaller antenna and is easy to deploy in mobile platforms, and it is sensitive to weak precipitation.

The process of establishing the new parameter is described in Section 2. Section 3 shows and discusses the typical vertical structure characteristics of the new parameter in terms of radial height indicator (RHI) data from X-band radar and the performance and simple application of the new parameter in S-band weather radar volume scan data. The limitations of the new parameter are discussed in Section 4. A summary is given in Section 5.

## 2. Axis Ratio Uniformity Index

### 2.1. Approximate Relationship between the Reflectivity Ratio, Dielectric Properties and Axial Ratio

In this section, an approximate relationship between the reflectivity ratio, dielectric properties and axial ratio is proposed for the derivation of the new parameter presented in Section 2.3. When the scattering amplitude of ellipsoidal particles is calculated using the Rayleigh–Gans formula [1], both the axis ratio of the particle ($r$) and the dielectric constant ($\varepsilon$) exist in a nonlinear form, and these two parameters are not easily separated to form independent product terms. In previous studies, dielectric parameters were often regarded as fixed values according to the phase state of the particles, and then the approximate relationship of other parameters was discussed. For example, $K_{DP}$ can be approximated as the product of the rain content, mass-weighted axial ratio of the raindrop, and a constant

term containing the given $\varepsilon$ [29]. However, for mixed phase clouds, the phase state of particles needs to be considered as a variable since they are not fully known in advance.

The dielectric property parameters of particles are one of the key parameters that determine the scattering ability of particles. These parameters are usually considered to be related to the material of the object, incident wavelength and ambient temperature. There are two equivalent descriptions of dielectric properties: the complex dielectric constant $\varepsilon = \varepsilon_r + i\varepsilon_i$ and the complex refractive index $m = m_r + im_i$, where $\varepsilon_r$ and $\varepsilon_i$ denote the real and imaginary parts of $\varepsilon$, $m_r$ and $m_i$ denote the real and imaginary parts of $m$, respectively, and $i^2 = -1$. The conversion relationships between $\varepsilon$ and $m$ are as follows:

$$m_r^2 = 0.5\left(\sqrt{\varepsilon_r^2 + \varepsilon_i^2} + \varepsilon_r\right) \tag{1}$$

$$m_i^2 = 0.5\left(\sqrt{\varepsilon_r^2 + \varepsilon_i^2} - \varepsilon_r\right) \tag{2}$$

For different hydrometeor phase states, the Ray scheme [30] is used to calculate the dielectric parameters $\varepsilon$ of pure water and pure ice. For an ice water mixture and spongy ice (mixture of ice and air), the overall dielectric constant is calculated according to the mass fraction method, and the Debye scheme [31] is selected as follows:

$$\frac{\varepsilon^{(mix)} - 1}{\varepsilon^{(mix)} + 2} = f \cdot \left(\frac{\varepsilon^{(1)} - 1}{\varepsilon^{(1)} + 2}\right) + (1 - f) \cdot \left(\frac{\varepsilon^{(2)} - 1}{\varepsilon^{(2)} + 2}\right) \tag{3}$$

where $\varepsilon^{(1)}$ and $\varepsilon^{(2)}$ are the complex dielectric constants of the two components, $\varepsilon^{(mix)}$ is the overall complex dielectric constant of the mixture, and $f$ is the volume fraction of the first component. The comparison of different schemes [31] shows that although the scheme of Equation (3) is not the most accurate scheme, the difference is very small compared with the best Maxwell–Garnett scheme, and the constraint conditions are the least accurate.

Some typical phases of hydrometeor particles in clouds and their corresponding dielectric properties are listed to find a simplified representation. The dielectric properties of pure water (clouds and raindrops) and pure ice (solid graupel and hail) are set according to the corresponding material. Mixtures of ice and water with $f = 0.5$ are used to characterize particles that are melting or freezing. Mixtures of ice and air (aggregated snow and ice crystals) with $f = 0.1$ and 0.5 are used to characterize spongy ice particles. For pure water, ambient temperatures of 0, 10, and 20 °C are selected to reflect the effect of temperature change on the dielectric properties. Since the dielectric properties of ice vary little with temperature, pure ice and other mixtures are set to 0 °C. Another problem is that $\varepsilon_r$ and $\varepsilon_i$ may vary differently with temperature, which leads to two variables of comparable magnitudes that need to be discussed. Note that $m_r$ is clearly larger than $m_i$ (Equations (1) and (2)); thus, only $m_r$ is taken as a dielectric characteristic parameter in the following attempts to characterize different phases.

Figure 1a gives $m_r$ at different phases and temperatures. The $m_r$ of water increases slightly with temperature in the X-band, while in the S-band, it decreases slightly or can be considered as undergoing little change. However, when water transitions to an ice/water mixture or ice, $m_r$ decreases, which generally has both nonlinear and nonmonotonic characteristics that are not easily used to form a simple model. However, if the reciprocal of $m_r$ is taken, it can be found that the $m_r^{-1}$ of spongy ice, pure ice, an ice/water mixture and pure water decrease somewhat linearly (Figure 1b). The different temperatures have little effect on $m_r^{-1}$ at this time. Therefore, $m_r^{-1}$ can be used as an available parameter to characterize the dielectric properties of the different phases.

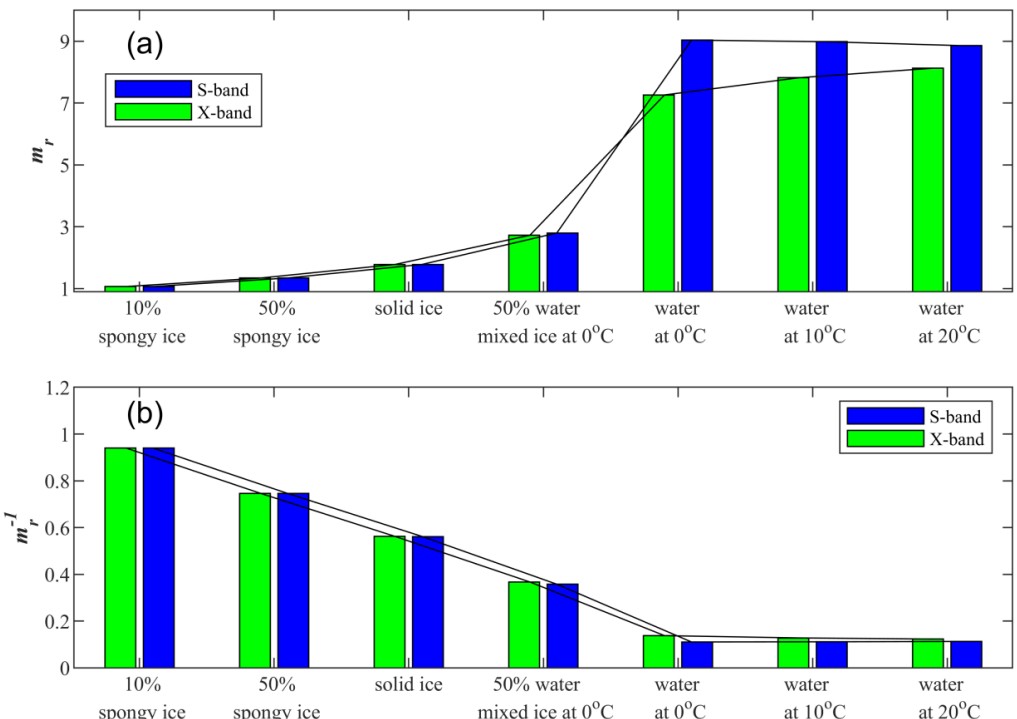

**Figure 1.** Dielectric property parameters: (**a**) $m_r$ and (**b**) $m_r{}^{-1}$ of hydrometeors at different phases and temperatures.

The reflectivity variables are calculated using the T-matrix method [32]. For raindrops, the relevant theoretical calculation schemes of reflectivity variables [1] are as follows:

$$Z_{h,v} = \frac{4\lambda^4}{\pi^4 |K_w|^2} \int_{D_{min}}^{D_{max}} |S_{hh,vv}(D)|^2 N(D) dD \tag{4}$$

$$Z_{H,V} = 10\log_{10} Z_{h,v} \tag{5}$$

$$Z_{dr} = \frac{Z_h}{Z_v} \tag{6}$$

$$Z_{DR} = 10\log_{10} Z_{dr} = Z_H - Z_V \tag{7}$$

where $Z_h$ (or $Z_v$), with lowercase subscripts, have linear units (mm$^6$/m$^3$), $Z_H$ (or $Z_V$), with uppercase subscripts, are in log units (dBZ), and H or V represent horizontal or vertical polarization, respectively. $Z_{DR}$ is in log units (dB). $Z_{dr}$ is the dimensionless reflectivity ratio. $K_w$ is associated with the dielectric property ($K_w = (\varepsilon - 1)/(\varepsilon + 2)$). $\lambda$ (unit: m) is the wavelength of the radar. $S_{hh,vv}$ is the backscattering amplitude of a single hydrometeor particle in a horizontal or vertical channel. $D$ is the equivalent spherical diameter of a particle, and $N(D)$ is the particle number concentration density. $D_{min}$ and $D_{max}$ are the lower and upper limits of the drop size distribution, respectively.

A common problem in the simulation of particle scattering properties is that there are many dimensions that can be discussed, such as phase, shape, axis ratio and size distribution. Here, an individual particle is first discussed, trying to find some available laws that are less affected by particle size. To avoid confusion with the $Z_{dr}$ of the particle group, the reflectivity ratio of a single particle is represented by the symbol $\eta_{dr}$. Taking an ellipsoidal particle with $r = 2$ as an example, the effect of different phases on $\eta_{dr}$ is analyzed (Figure 2). Note that $r$ here is defined as the ratio of the horizontal scale to the vertical scale of the particle relative to the polarization direction of the radar beam, which is contrary to the definition of the raindrop axis ratio used in previous studies [33–36]. Figure 2 shows that $\eta_{dr}$ increases as the phase of the particles becomes closer to pure water. For $D$ greater than

approximately 5 mm (in the S-band) and 2.5 mm (in the X-band), the $\eta_{dr}$ of the pure water particle shows large oscillations due to the effect of Mie scattering. For the mixed phase, $\eta_{dr}$ increases slightly with $D$. However, the general rule is still that the phase corresponding to $m_r^{-1}$ can amplify the value of $\eta_{dr}$. Therefore, we can use this relationship between the phase and $\eta_{dr}$ of a single particle as an available approximation and the Mie scattering effect caused by the change in $D$ as a potential error factor leading to the inaccuracy of this approximation. For example, based on this approximation, raindrops with a particle size greater than 6 mm in the S-band will introduce uncertainty, while particles of 3 to 4 mm in the X-band will lead to overestimation of the axial ratio. Considering that the actual radar detection variable is an integral of a group of particles, the above error factors will not always dominate.

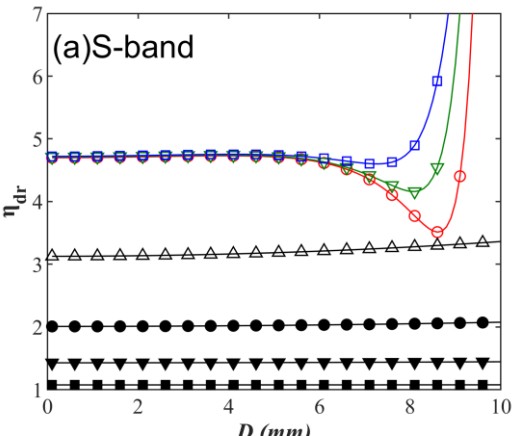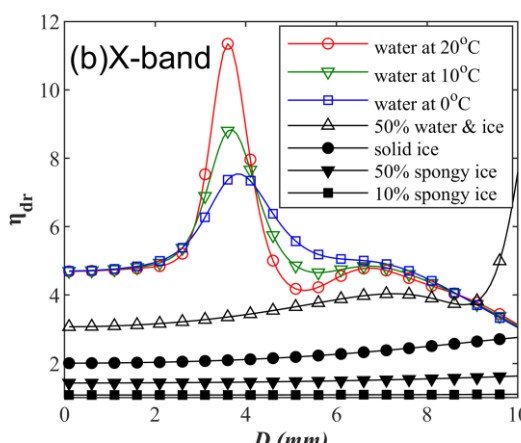

**Figure 2.** Variation of $\eta_{dr}$ with the spherical equivalent diameter $D$ for a single ellipsoid water particle with an axial ratio $r = 2$ under different phase conditions. (**a**): S-band, (**b**): X-band. $\eta_{dr}$ represents the $Z_{dr}$ of a single particle.

Then, if there is an available relationship with $\eta_{dr}$, phase and $r$ can be discussed by ignoring the effect of $D$ on $\eta_{dr}$, provided that the $r$ is fixed, and $D$ is fixed to 1 mm in the simulations. Figure 3 shows that the contribution of $r$ also amplifies $\eta_{dr}$. However, this is not easily applied since $\eta_{dr}$ changes along both $r$ and phase. Hence, an approximate significant linear relationship is proposed here, taking $X = m_r^{-1}$ as an independent variable and $Y = (\eta_{dr}^{0.5})/(r - 1)$ as a dependent variable to form the linear regression $Y = a_1 X + a_0$ (Figure 4).

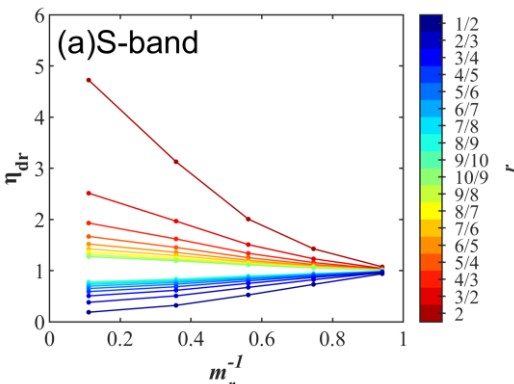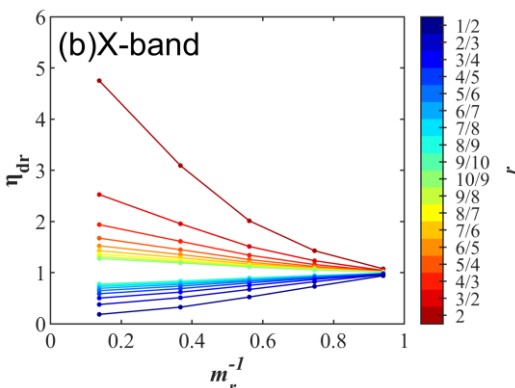

**Figure 3.** Variation of $\eta_{dr}$ with $m_r^{-1}$ for a single ellipsoidal water particle with different axial ratios (colors of the lines) when $D = 1$ mm. (**a**) S-band, (**b**) X-band.

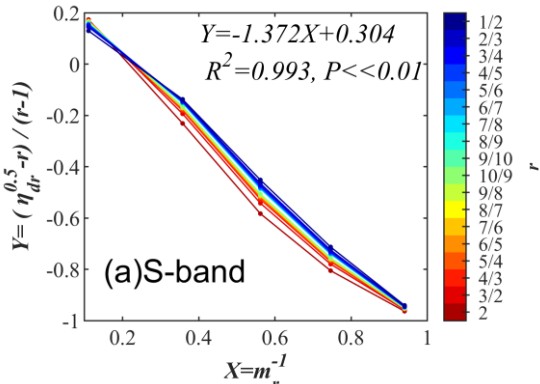 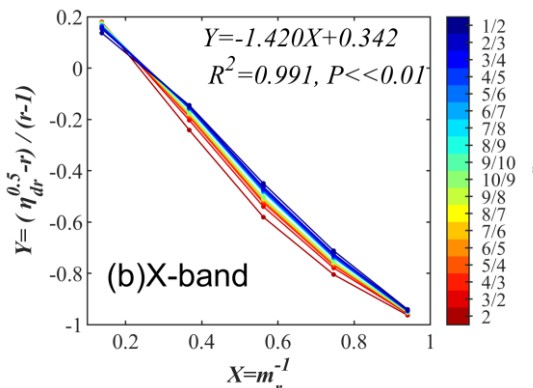

**Figure 4.** Linear approximation between $m_r$, $\eta_{dr}$ and $r$ of a single ellipsoidal particle. ($R^2$ is the goodness of fit, and $P$ is the significance of the linear fit. (**a**) S-band, (**b**) X-band.

According to the slope $a_1$ and intercept $a_0$ obtained by the linear fit shown in Figure 4, a new particle phase parameter $E$ is defined instead of $m_r$ to characterize the phase state:

$$E = a_1 / m_r + a_0 \tag{8}$$

Furthermore, for a single ellipsoidal particle, a simple relationship between $\eta_{dr}$, $E$ and $r$ can be obtained, as shown in Equation (9):

$$\sqrt{\eta_{dr}} = E \cdot (r - 1) + r \tag{9}$$

Three points also need to be noted:

(1) The approximate linear relationship in Figure 4 is not the most accurate approximation. If $r$ is logarithmic, an approximation with less error can be constructed. However, the approximate linear relationship shown in Figure 4 and Equation (9) is now more readily available for deriving the new index presented below. Therefore, Equation (9) is still used.

(2) The process of eliminating $E$ will be shown in Section 2.3; thus, the values of $E$, $a_1$, and $a_0$ in Equation (8) are no longer discussed in the following sections.

(3) The effects of radar scanning elevation, particle orientation and nonellipsoidal shape are not considered here. Therefore, $r$ should be considered as the flattening or narrowing of the particle in the horizontal and vertical polarization directions as detected by the radar.

### 2.2. Approximate Relationship between $\rho_{hv}$ and Reflectivity

In this section, an approximate relationship between $\rho_{hv}$ and the reflectivity variables is proposed for the derivation of the new parameter presented in Section 2.3. According to the principles involved in dual polarization radar detection, $\rho_{hv}$ itself reflects the uniformity of the particle shape in the detection volume, but it is also affected by the radar observation system and noise [37,38]. Referring to the basic definition of $\rho_{hv}$, the ideal $\rho_{hv}$ formula [3] is shown in Equation (10):

$$\rho_{hv}^{(Ideal)} = \frac{\langle S_{hh}^* S_{vv} \rangle}{\sqrt{\langle |S_{hh}|^2 \rangle \langle |S_{vv}|^2 \rangle}} \tag{10}$$

where <...> represents the volumetric average. The $S_{hh,vv}$ values are complex, and the molecular component included in Equation (10) requires conjugate multiplication, which makes it difficult to establish numerical connections with known parameters. For this reason, an approximate $\rho_{hv}$ is proposed here as Equation (11):

$$\rho_{hv}^{(Approx)} = \frac{\sum \left( \sqrt{b_h} \cdot \sqrt{b_v} \right)}{\sqrt{\sum b_h \cdot \sum b_v}} \tag{11}$$

where $b_h$ and $b_v$ reflect the contribution of a single particle to $Z_h$ and $Z_v$, respectively, as follows:

$$\sum b_h = Z_h, \sum b_v = Z_v \tag{12}$$

In this way, the relationship between $\rho_{hv}$ and the reflectivity of a single particle is established.

To verify the hypothetical approximation of Equation (11), a particle size distribution is necessary since $\rho_{hv}$ is based on a particle group and is calculated by an integral or volumetric average. The range of the axial ratio and size distribution of ice particles may be too large and random, which is not easily resolved in a representative enumeration study; thus, a simpler raindrop size distribution (RSD) model is taken. By enumerating some typical RSDs, the difference between the ideal and approximate $\rho_{hv}$ in different cases can be compared. A common RSD model can be expressed by a gamma distribution with three parameters [1] as follows:

$$N(D) = N_T \frac{(3.67 + \mu)^{\mu+1}}{\Gamma(\mu+1)D_0} \left(\frac{D}{D_0}\right)^{\mu} e^{[-(3.67+\mu)\frac{D}{D_0}]} \tag{13}$$

where $N_T$ is the number concentration of particles, $D_0$ is the spherical equivalent volume median diameter, and $\mu$ is the shape parameter of the RSD. When considering the "shape size" ($D$ with $r$) model of raindrops, since the trends of D with $r$ are not much different in previous models using fixed parameters [33,35,36], the scheme presented in [36] is taken as a typical case. Another key issue is the setting of $D_{max}$. In common rainfall, raindrops larger than 6 mm are rare, but in severe convective rainfall, large raindrops of approximately 10 mm are often observed. Choosing a different $D_{max}$ may result in a large difference in the variables after integration according to RSD, thus $D_{max}$ values of 6 and 10 mm are both taken to represent common and typical severe rainfall cases, respectively. D is from 0.1 to $D_{max}$ with a 0.1 mm interval.

When enumerating different sets of RSD parameters, $N_T$ is not enumerated since $\rho_{hv}$ does not involve the absolute number of particles. $D_0$ starts from 0.1 mm with a 0.1 interval, and its upper limit is determined according to the constraint relation $D_{max}/D_0 \geq 2.5$ [39] to limit the truncation error. $\mu$ is from $-0.8$ to 16 with a 0.2 interval to represent exponential shape ($\mu \leq 0$) and single peak shape (larger $\mu$) distributions. By the combination of different $D_{max}$, $D_0$ and $\mu$, RSD parameters in wide value ranges are enumerated to cover possible real conditions of raindrops.

The difference between the approximate $\rho_{hv}$ and the ideal $\rho_{hv}$ is evaluated by common statistics, including the correlation coefficient (R), mean absolute error (MAE) and mean relative error (MRE, see Appendix A for definitions). The results are shown in Figure 5. For general rainfall in the S-band (Figure 5a), the approximate $\rho_{hv}$ can be considered to be consistent with the ideal $\rho_{hv}$. In other cases, the approximate $\rho_h$ is larger than the ideal $\rho_h$, but the deviation is generally limited. The largest deviation appears in the case of severe rainfall in the X-band (Figure 5b), but the MRE is only 0.13%. Therefore, the approximate $\rho_{hv}$ in Equation (11) is considered basically consistent with the ideal $\rho_{hv}$ for rainfall cases.

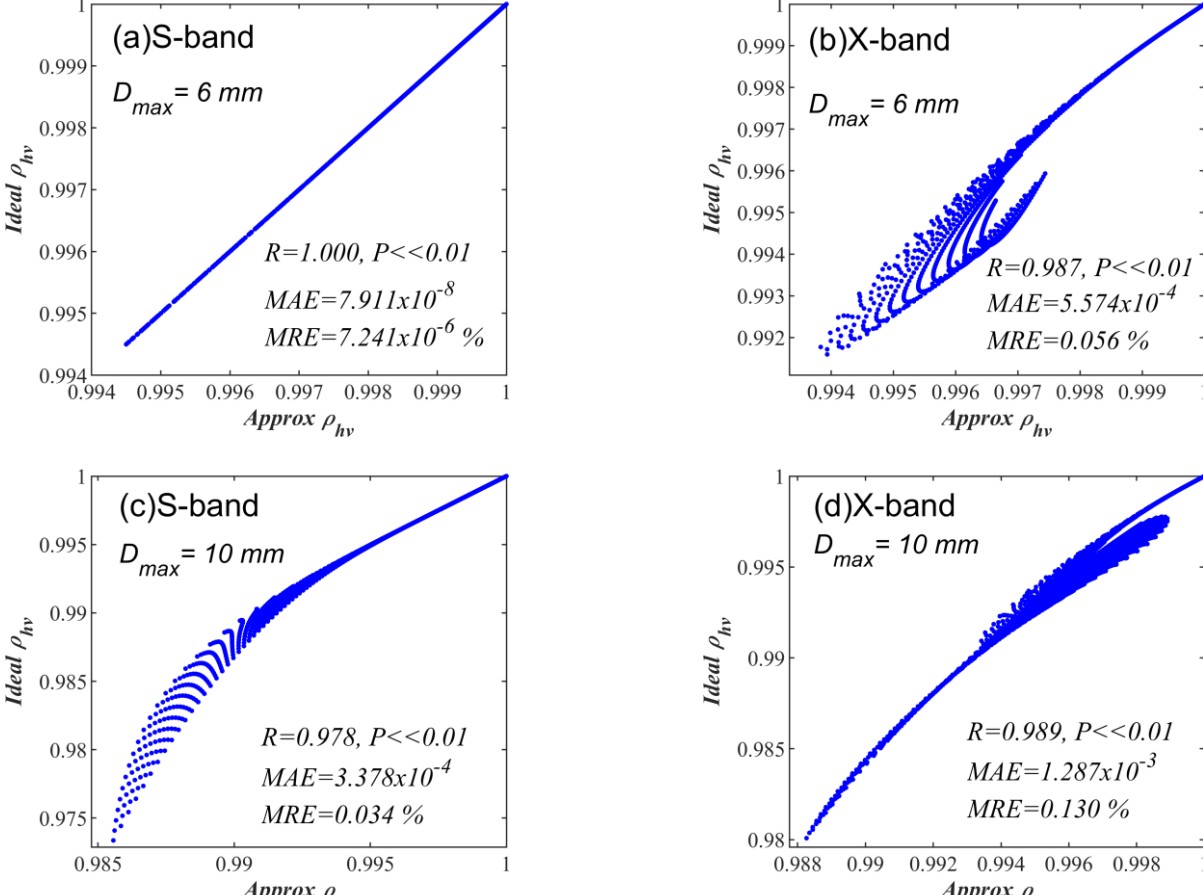

**Figure 5.** Relationships between approximate $\rho_{hv}$ in this study and ideal $\rho_{hv}$ in raindrops. (The values of $D_0$ are from 0.1 mm to $D_{max}$ in 0.1 mm intervals. The values of μ range from −0.8 to 16 in intervals of 0.2. R is the correlation coefficient between the approximate $\rho_{hv}$ and ideal $\rho_{hv}$ enumerated samples, and *P* is the significance of R. MAE is the mean absolute error, and MRE is the mean relative error. See Appendix A). (**a**) S-band and $D_{max}$ = 6 mm, (**b**) X-band and $D_{max}$ = 6 mm, (**c**) S-band and $D_{max}$ = 10 mm, (**d**) X-band and $D_{max}$ = 10 mm.

### 2.3. Construction of the New Parameter

In this section, the derivation of the new parameter is proposed based on the approximations obtained in the previous two sections. First, according to the provisions of $b_h$ and $b_v$ (Equation (12)), Equation (9) can be transformed into Equations (14) and (15):

$$\frac{\sqrt{b_h}}{\sqrt{b_v}} = E \cdot (r-1) + r \qquad (14)$$

$$\sqrt{b_h} = (E \cdot (r-1) + r) \cdot \sqrt{b_v} \qquad (15)$$

The above still applies to a single particle. When considering an integrated particle group, a new weighted average axial ratio is defined as Equation (16):

$$\bar{r} = \frac{\sum (b_v \cdot r)}{\sum b_v} \qquad (16)$$

The $\bar{r}$ here is actually the "vertical reflectivity weighted average axial ratio" and can be considered to reflect the overall average axial ratio of the particle group. Then, combining

Equations (9) and (14) and the definition mode of Equation (16), $\overline{r^2}$ is further defined as follows:

$$\overline{r^2} = \frac{\sum(b_v \cdot r^2)}{\sum b_v} \tag{17}$$

Then, the $Z_{dr}$ of a particle group can be written as a relation of E, $\overline{r}$ and $\overline{r^2}$ as Equation (18):

$$Z_{dr} = \frac{\sum b_h}{\sum b_v} \tag{18}$$

$$= \frac{\sum\left[b_v \cdot (E \cdot r - E + r)^2\right]}{\sum b_v} \tag{19}$$

$$\approx (E+1)^2 \cdot \overline{r^2} - 2E \cdot (E+1) \cdot \overline{r} + E^2 \tag{20}$$

The *E* here is eventually moved outside the summation sign from Equations (19) to (20), where it represents the general phase of a particle group (which may be a mixture) and ignores the differences caused by the different phases of each particle. Furthermore, combining Equations (11), (14) and (18), $Z_{dr}$ and $\rho_{hv}$ can be written as the relation between *E* and $\overline{r}$, as shown in Equation (19):

$$\rho_{hv}\sqrt{Z_{dr}} = \frac{\sum[b_v \cdot (E \cdot r - E + r)]}{\sum b_v} \tag{21}$$

$$\approx E \cdot \overline{r} - E + \overline{r} \tag{22}$$

Thus far, there are two radar variables ($Z_{dr}$ and $\rho_{hv}$) that are used, while there are three unknowns: $E$, $\overline{r}$ and $\overline{r^2}$. Although absolute quantities such as total concentration and water content are avoided, there is still no way to solve all unknowns. To this end, a solution is proposed here that eliminates *E* by a combination of $Z_{dr}$ and $\rho_{hv}$ to finally obtain a relationship between $\overline{r}$ and $\overline{r^2}$:

$$\frac{\left(\rho_{hv}\sqrt{Z_{dr}} - 1\right)^2}{Z_{dr} - 2\rho_{hv} \cdot \sqrt{Z_{dr}} + 1} = \frac{(E+1)^2 \cdot (\overline{r} - 1)^2}{(E+1)^2 \cdot \overline{(r-1)^2}} \tag{23}$$

$$= \frac{(\overline{r} - 1)^2}{\overline{(r-1)^2}} \tag{24}$$

Note that Equation (24) can reflect the uniformity of *r* relative to 1 for a particle group. For example, the shape and orientation of ice particles may differ greatly, resulting in a large denominator and small molecule component in Equation (24), which eventually leads to the value of Equation (24) being close to 0. However, for raindrops, *r* is greater than 1 for most particles. This results in a value of Equation (24) between 0 and 1 and close to 1. In addition, the elimination of *E* between Equations (23) and (24) is equivalent to the elimination of the impact of phase on amplifying $\eta_{dr}$ or $Z_{dr}$. Finally, only one descriptive quantity for the axial ratio distribution uniformity of the particle group is obtained, and it is named the "axis ratio uniformity index" ($U_{ar}$):

$$U_{ar} = \frac{\left(\rho_{hv}\sqrt{Z_{dr}} - 1\right)^2}{Z_{dr} - 2\rho_{hv} \cdot \sqrt{Z_{dr}} + 1} \tag{25}$$

where the dimensionless $Z_{dr}$ can be transformed by $Z_{dr} = 10^{ZDR}$, and $Z_{DR}$ (in dB) is observed by radar. Therefore, $U_{ar}$ is a variable that can be calculated from radar measurements $Z_{DR}$ and $\rho_{hv}$.

The numerical distribution of $U_{ar}$ is shown in Figure 6, which is based on the same enumerated ranges of RSD as in Section 2.2. In the S-band, $U_{ar}$ values are mostly con-

centrated between 0.8 and 0.9, mostly above 0.7. In the X-band, the value distribution of $U_{ar}$ is more dispersed, but most values are more than 0.6. Less than 10% of the data have $U_{ar}$ values less than 0.1, and the relationship between $U_{ar}$, $D_0$, and $\mu$ is further examined (Figure 7). $U_{ar}$ rapidly decreased to 0 when $D_0$ was less than 0.5 mm, i.e., indicating that $U_{ar}$ does not have the ability to distinguish particles in the ice phase from raindrop groups with small particle diameters. However, since the case in which $D_0$ is less than 0.5 mm rarely appears in previous joint observation and retrieval studies based on weather radar and RSD [29,40,41], it can be considered that $U_{ar}$ can show a value close to 1 for raindrops, which is obviously different from the value close to 0 for most ice phase particles.

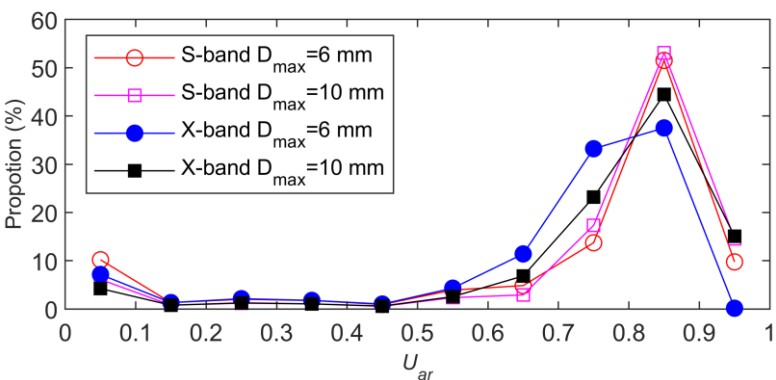

**Figure 6.** Distribution of $U_{ar}$ within the enumeration range of the RSD parameters. (The values of $D_0$ and $\mu$ are the same as those in Figure 5).

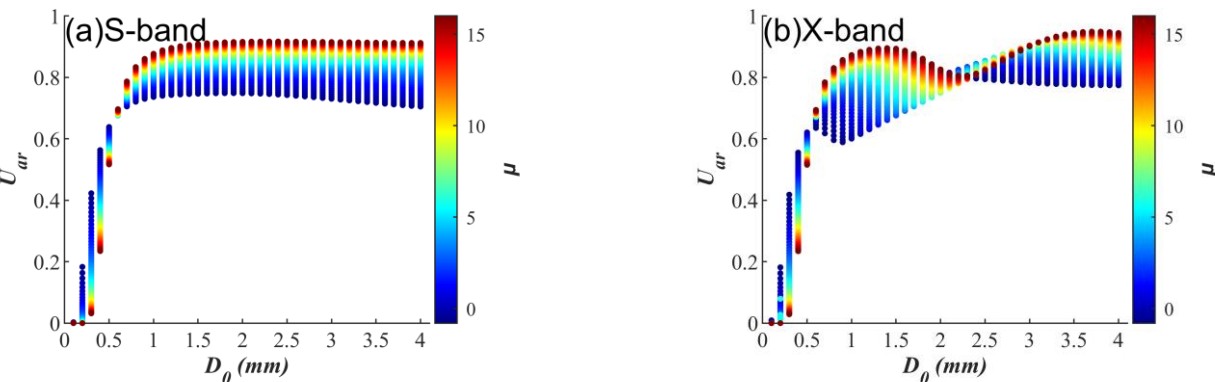

**Figure 7.** Relationships of $U_{ar}$ versus $D_0$ and $\mu$ when $D_{max}$=10 mm. ($D_{max}$ = 10 mm. The values of $D_0$ and $\mu$ are the same as those in Figure 5). (**a**) S-band, (**b**) X-band.

In summary, $U_{ar}$ has the potential to identify rain areas. However, the $U_{ar}$ probability distribution shown in Figure 6 is not the probability distribution considered for actual detection, and none of the above discussions include noise in $\rho_{hv}$. The practical application effect of $U_{ar}$ will be discussed below.

## 3. Performance of $U_{ar}$ on Real Observations

### 3.1. Typical Features of Vertical Structures of $U_{ar}$ on X-Band RHI Radar Data

3.1.1. Overview of RHI Data during a Convective Event

In this section, the vertical distribution characteristics of $U_{ar}$ are discussed using RHI data obtained by X-band dual polarization radar. The selected case is a convective event in Beijing that occurred during 18:00–19:30 (Local Standard Time (LST), GMT+8) on 7 September 2016. The radar is a 714XDP-A type X-band dual polarization mobile radar belonging to the Key Laboratory of Cloud-Precipitation Physics and Severe Storms (LACS), Institute of Atmospheric Physics (IAP), Chinese Academy of Sciences (CAS). That

radar was deployed for field observations at the Beijing Olympic Water Park (116.68°E, 40.18°N) during the summer from 2015 to 2019. The main characteristics of this radar are listed in Table 1. Other information, including quality control and attenuation correction methods, can be seen in [42,43]. The vertical temperature profile used for the analysis was derived from the neighboring sounding station at 08:00 LST (station number: 54511, 116.28°E, 39.93°N). For the observation mode, plan position indicator (PPI) and RHI scans were switched manually in these observations. The PPI at 4° elevation was scanned first, followed by an RHI scan that aimed at the strong convection center, forming a cycle to track the evolution of the vertical structure of the severe convective cell. Figure 8 shows the convective system to the southwest of the radar moving southeastward.

**Table 1.** Characteristics of the X-band dual polarization radar used in this paper.

| Attribute | Value | Attribute | Value |
| --- | --- | --- | --- |
| Antenna diameter | 2.4 m | Linear dynamic range | >90 dB |
| Frequency | 9.37 GHz | Beam width | 1° |
| Antenna gain | 41.6 dB | Radial resolution | 150 m |
| Peak power | 80 kW | Observation range | 150 km |
| Polarization | Horizontal/Vertical | | |
| Pulse width | 0.5/1/2 μs | Elevation resolution in RHI mode | 0.17° |

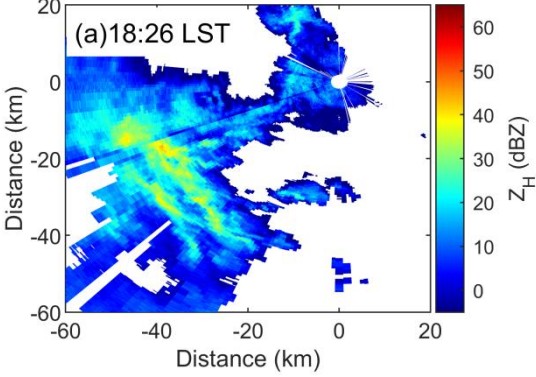 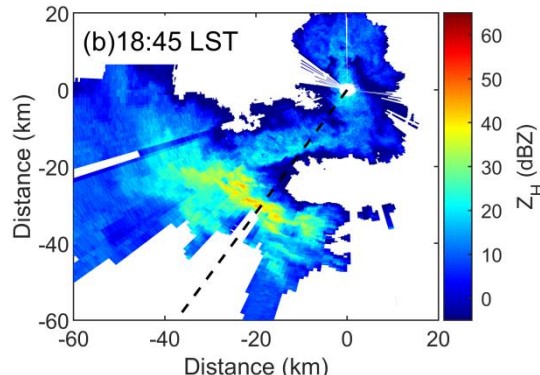

**Figure 8.** Sample PPI observations of $Z_H$. (The elevation of the PPI is 4°. The azimuth of the dashed line in Figure 8b is 212°. The data collection time is 7 September 2016). (**a**) 18:26 LST, (**b**) 18:45 LST.

Figure 9a–d show radar data at the RHI of 212° azimuth. The $K_{DP}$ is not shown here since it is not discussed in this paper. There is a stratiform cloud area from 0 to 25 km in the horizontal direction, and the BB characteristics of the ML are found below the 0 °C layer among $Z_H$, $Z_{DR}$ and $\rho_{hv}$. The ML, at a horizontal distance of approximately 20 km, has a sinking feature, which may be related to the rapid fall caused by the riming or coalescence of snow [3,27,28]. In the convective cloud region, the convective core is located at a horizontal distance of 35 km (Figure 9a), and there is a distinct $Z_{DR}$ column [44] feature (Figure 9b). The value of $\rho_{hv}$ in the lower layer and 40 km away is less than 0.4 (Figure 9c), which should be attributed to invalid observations caused by attenuation and noise. The corresponding $Z_H$ and $Z_{DR}$ values at such areas are automatically masked during the quality control process, despite some of the $Z_{DR}$ values remaining abnormally low at the rear of the radar beams. The Doppler radial velocity ($V_R$, Figure 9d) shows convergence below 6 km at the convective core, while divergence appears at a height of approximately 10 km, indicating a strong updraft. Figure 9e shows the $U_{ar}$ proposed in this paper. In addition, a hydrometeor classification (HC) result is shown in Figure 9f as a reference for the particle phase, using the scheme from the work of Feng et al. [45], which is an ensemble and improved version from previous studies [9–12].

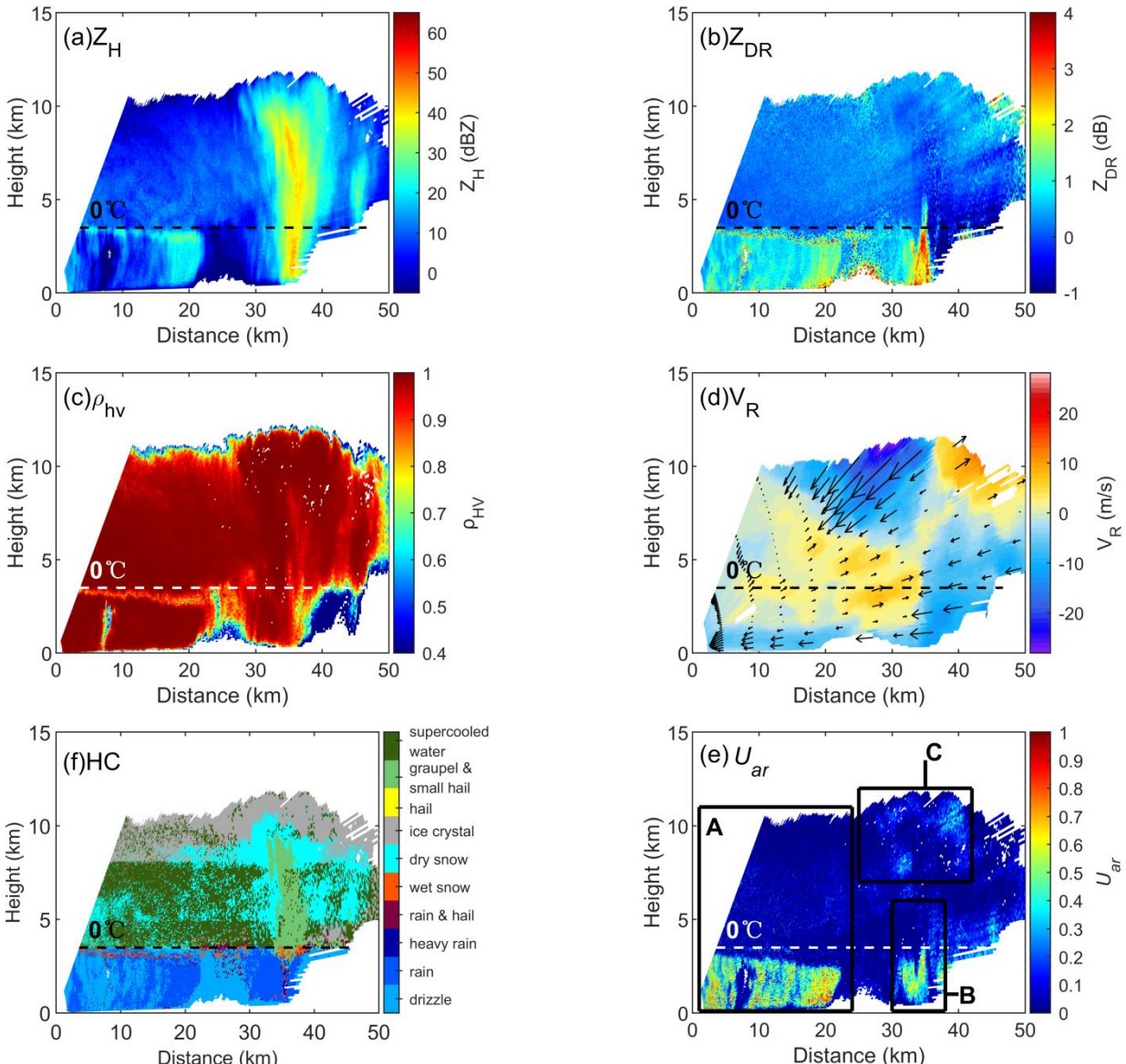

**Figure 9.** Sample RHI observations and retrievals of the convective system from the X-band dual polarization weather radar. (The time is 18:46:30 LST. The azimuth is the same as the dashed line in Figure 8b. The maximum elevation is 44°). (**a**) $Z_H$, (**b**) $Z_{DR}$, (**c**) $\rho_{hv}$, (**d**) $V_R$, (**f**) HC, (**e**) $U_{ar}$.

The following focuses on the three regions shown in Figure 9e, where region A is a stratiform cloud area, region B is the lower area of the convective cloud, and region C is the upper area of the convective cloud.

In region A, $U_{ar}$ suddenly appears as a whole layer with values larger than approximately 0.5 from a certain distance below the 0 °C layer to the ground (Figure 9e), while the HC result shows wet snow corresponding to the ML, and then rain and drizzle below. These results show that HC and $U_{ar}$ are basically consistent in terms of identifying rainy areas. Moreover, the top of the high-value area of $U_{ar}$ exhibits a sinking feature consistent with ML.

In region B, there is an obvious difference between HC and $U_{ar}$. The high value areas of $U_{ar}$ show a clear 'U'-shaped spatial distribution, while HC shows a very narrow range of heavy rain, graples and hail mixing with rain in other locations.

In region C, $U_{ar}$ has a high value area with the maximum value approaching 0.4 at the divergence area of $V_R$, corresponding to dry snow in the HC results. This phenomenon may be explained by the fact that strong horizontal winds contributed to the formation of high-level snow clustering and the maintenance of quasi-horizontal orientation. However,

since the spatial scope of that feature is limited, the value of $U_{ar}$ is not so large, and ice habits are not a focus of this study, there is no further discussion of it in the following sections. In Sections 3.1.2 and 3.1.3 below, only areas A and B are discussed.

3.1.2. Analysis of the Stratiform Cloud Area

Average vertical profiles of polarimetric variables in region A are counted and shown in Figure 10. There are notable ML features with extreme values and strong gradients, where $Z_H$ has a peak and $\rho_{hv}$ has a valley from the 0 °C level (3.48 km) down to approximately 9 °C (2.37 km). $Z_{DR}$ also shows a small peak in this ML. However, $U_{ar}$ is close to 0 in the cold cloud area and changes in the ML, mutating into an average of more than 0.4 below the ML. This result apparently shows that $U_{ar}$ is a step-like mutation signal in a three-layered cold cloud, which is unlike conventional variables (such as $Z_H$, $Z_{DR}$ and $\rho_{hv}$) that exhibit extreme values in ML.

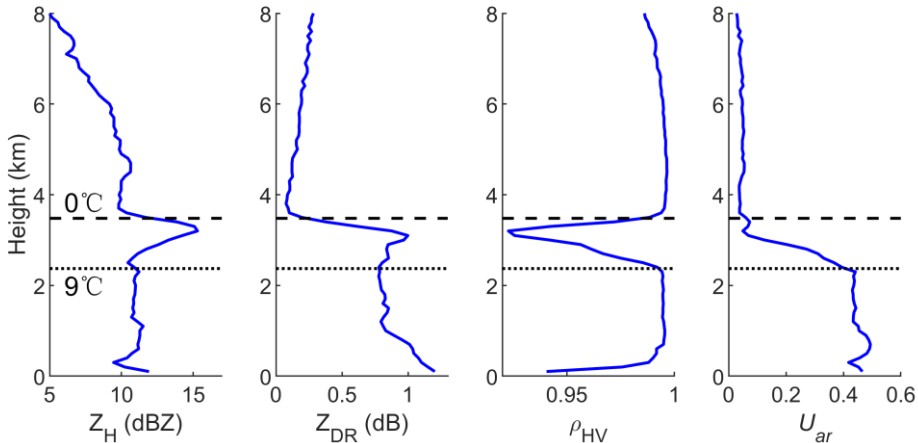

**Figure 10.** Mean vertical profile in area A. (Statistics by data points with $Z_H > 0$ dBZ).

The threshold value of $U_{ar}$ used for determining the raindrop areas should be given according to the $U_a$ probability distribution statistics below the ML. If the data below the ML where $Z_H$ is larger than 0 dBZ are counted (Figure 11a), $U_{ar}$ is concentrated between 0.4 and 0.5, but there is a certain distribution from 0 to 1 that makes it difficult to select a threshold. If the statistics are performed in areas with a slightly stronger reflectivity, such as 20 dBZ, which is generally considered to have obvious rainfall, $U_{ar}$ values range, at most, from 0.6 to 0.7 (Figure 11b), and more than 90% are above 0.4. Therefore, it is possible to identify raindrop areas easily only by setting a $U_{ar}$ threshold and without temperature input instead of determining the boundaries of ML first. Further quantitative analysis of the threshold is presented in Section 3.3.

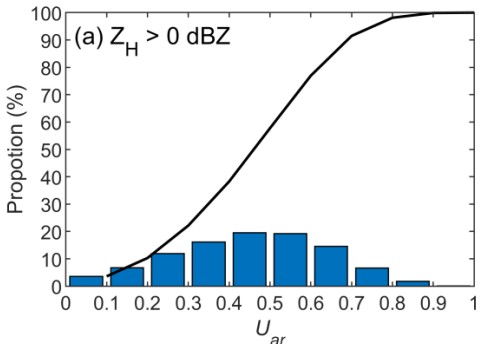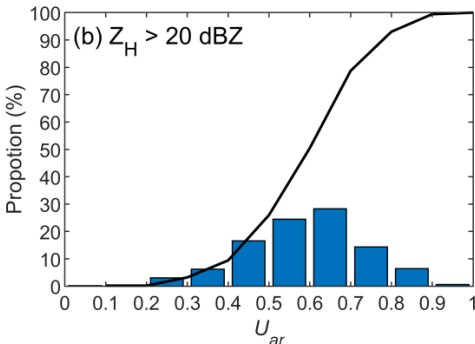

**Figure 11.** Proportions of $U_{ar}$ values in the warm layer (below 2.37 km, warmer than 9 °C) of area A: (**a**) points with $Z_H > 0$ dBZ and (**b**) points with $Z_H > 20$ dBZ. The bars are the proportions within given ranges, and the black lines are the accumulated proportions.

### 3.1.3. Analysis of a Convective Cloud Area in the Lower Levels

Figure 12 shows the enlarged vertical structure of variables in region B. The divergence of $V_R$ is added (Figure 12c) to better diagnose the distribution of vertical airflow in the cloud, which is defined as $dV_R/ds$, where s is the radial distance of a radar beam. Strong convergence ($dV_R/ds < 0$) extends from the ground to a height of 5 km and reaches a maximum above the 0 °C layer, which indicates a deep and strong updraft here that is consistent with the position of the $Z_{DR}$ column shown in Figure 12b. The $Z_{DR}$ column is a phenomenon in which the high $Z_{DR}$ region extends above the 0 °C height [45] and is thought to be closely related to the transport of large raindrops and strong updrafts in the supercooled layer [46–50]. The right branch of the U-shape of $U_{ar}$ corresponds to the $Z_{DR}$ column and the strong convergence of $V_R$, which indicates that the corresponding raindrops are transported upward by the updraft. The upper bound of the right branch of that U-shape crosses the 0 °C layer, indicating that the raindrops freeze after being transported to the supercooled layer. The left branch of the U-shape is 500 m lower than the right branch and is deflected away from the strong convergence area, which can be inferred as raindrops formed by the falling and melting of high-level ice particles.

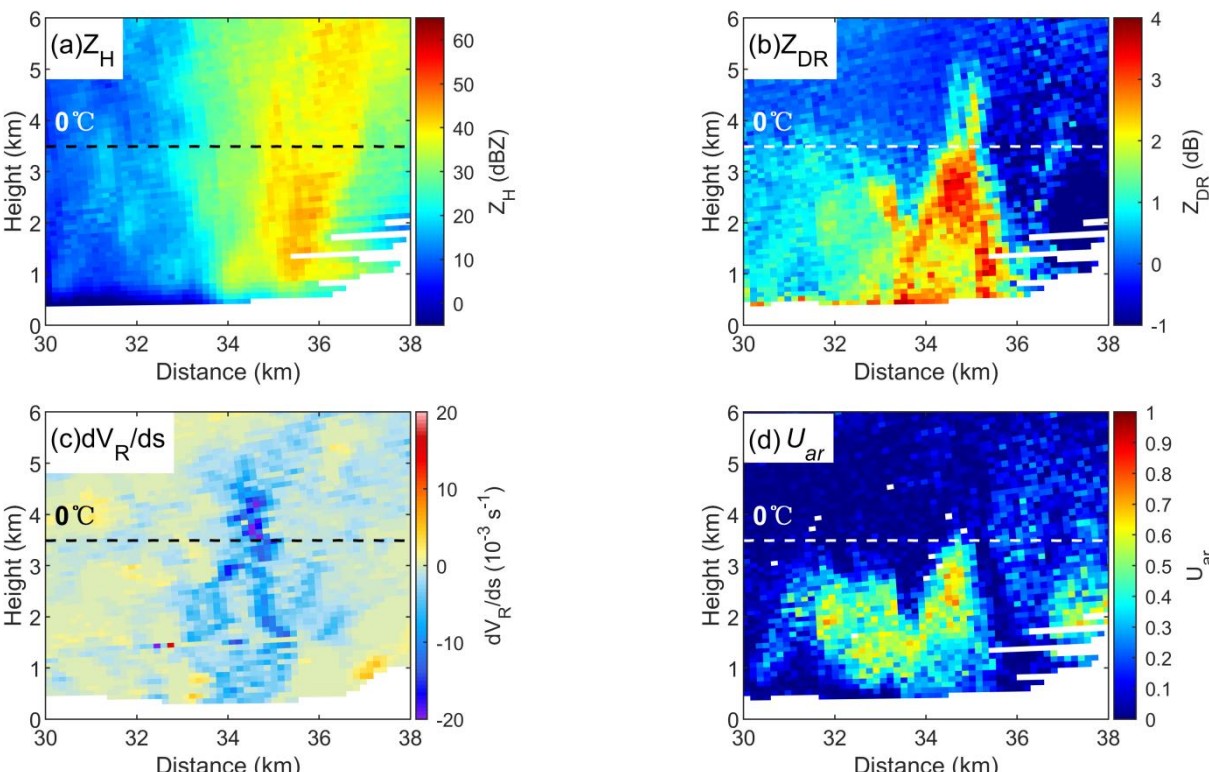

**Figure 12.** Variables in area B with the $Z_{DR}$ column. (**a**) $Z_H$, (**b**) $Z_{DR}$, (**c**) $dV_R/ds$, (**d**) $U_{ar}$.

In the analysis of the above features that change with time, due to the large amount of low-level occlusion during the period preceding what is shown in Figures 9 and 12, a later time is selected to track the change in the convective cell. Figure 13 shows PPI data 7 min later and RHI data 9 min later, where the RHI is obtained by tracking the horizontal movement of the $Z_H$ core. Figure 14 shows an enlarged view of the convective core at a low level, which is similar to Figure 12. The divergence of $V_R$ ($dV_R/ds > 0$, Figure 14c) below 1 km indicates the dominating downdraft caused by rainfall. The $Z_{DR}$ column no longer exists (Figure 14b), which may be the result of deep updrafts disappearing. At the same time, $U_{ar}$ no longer displays a U-shape (Figure 14d). The upper bound of the large value of $U_{ar}$ is approximately 1 km below the 0 °C layer, showing the characteristics of large-scale melting of ice particles into raindrops. In summary, it can be inferred that the left and right branches

of the U-shaped $U_{ar}$ in the lower troposphere correspond to rain formed by the melting of ice particles and rain transported upward by the updraft, respectively. This set of processes is very similar to Conway and Zrnic's explanation of the formation mechanism of the $Z_{DR}$ column [46]. Therefore, the study presented here can not only be used as evidence to support previous studies but can also expand the means of future research on the $Z_{DR}$ column.

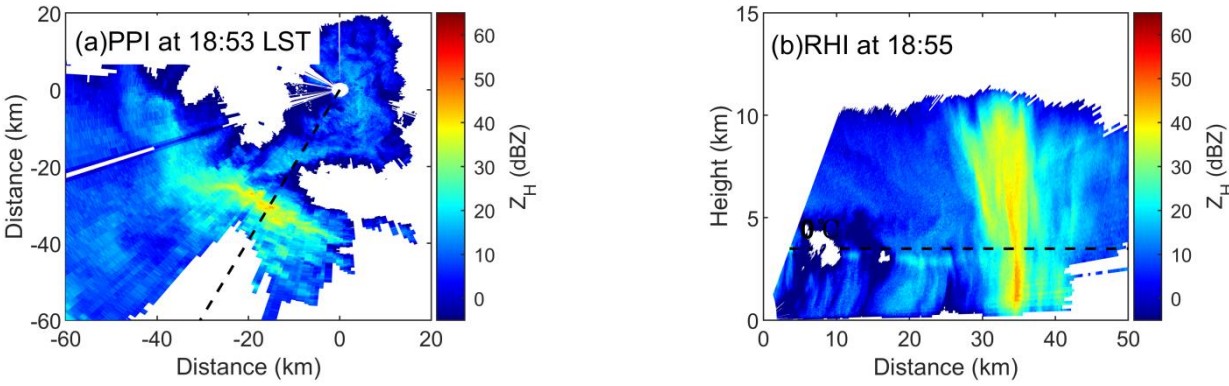

**Figure 13.** PPI and RHI of $Z_H$ at subsequent time intervals. (**a**) PPI at 18:53 LST, (**b**) RHI at 18:55.

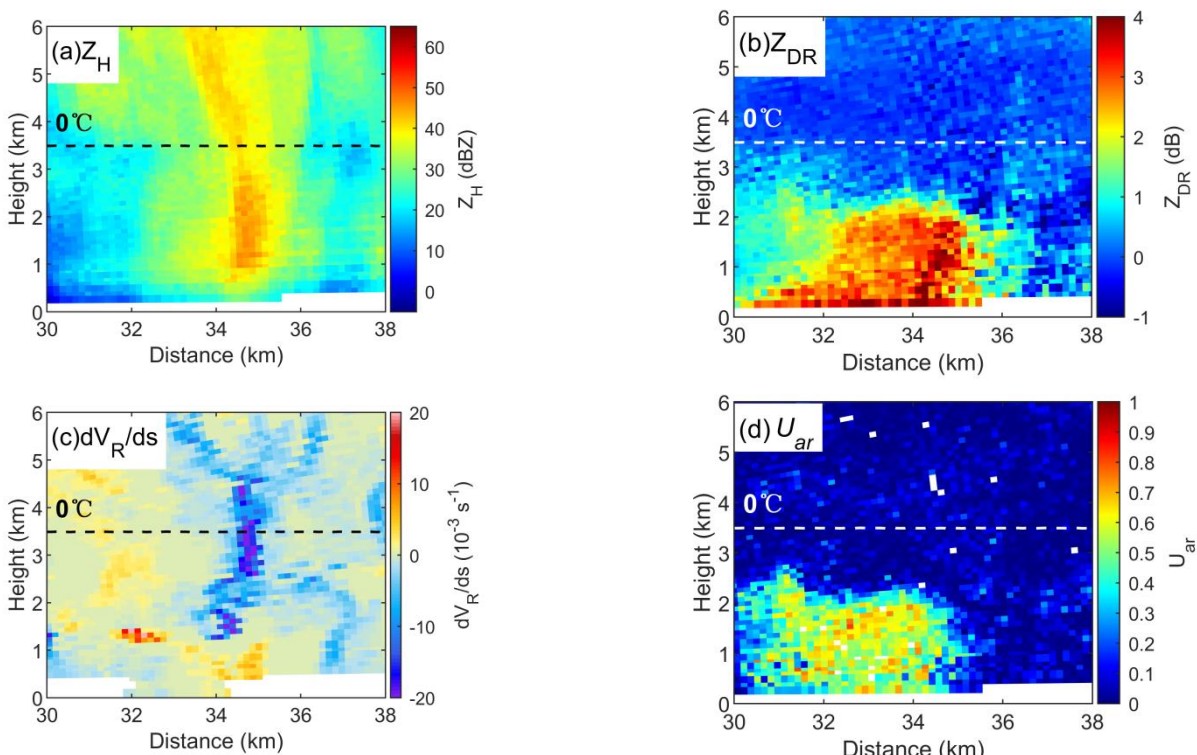

**Figure 14.** Variables after the $Z_{DR}$ column disappeared. (**a**) $Z_H$, (**b**) $Z_{DR}$, (**c**) $dV_R/ds$, (**d**) $U_{ar}$.

## 3.2. Performance of $U_{ar}$ on S-Band Volume Scans Radar Data

In this section, S-band operational dual polarization radar volume scan data are used to evaluate the application of $U_{ar}$. The involved case is a severe convective event that occurred in Shandong Province, China, during the evening on 17 May 2020. The radar is a dual polarization radar upgraded from CINRAD-SA type radar (station number: Z9532, 120.23°E, 35.99°N). The main characteristics of this radar are listed in Table 2. This radar performs a volume scan containing nine elevations from 0.5° to 19.5° (commonly called VCP-21 mode) in approximately 6 min. The vertical temperature profile used for the

analysis was derived from the neighboring sounding station at 20:00 LST (station number: 54857, 120.33°E, 36.06°N).

**Table 2.** Characteristics of the S-band dual polarization radar used in this paper.

| Attribute | Value | Attribute | Value |
|---|---|---|---|
| Antenna diameter | 8.5 m | Pulse width | 1.57/4.57 μs |
| Frequency | 2.88 GHz | Linear dynamic range | >85 dB |
| Antenna gain | >45 dB | Beam width | 0.93° |
| Peak power | 650 kW | Radial resolution | 250 m |
| Polarization | Horizontal/Vertical | Observation range | 460 km |

The mid-late stage data regarding the convective system development are selected, where there is a large range of stratiform cloud areas behind the convection line. PPI data with obvious ML features are shown in Figure 15. The BB signal in $Z_H$ is not obvious (Figure 15a), while $Z_{DR}$ and $\rho_{hv}$ both have a ring area with extreme values and rough texture, indicating that the ML is between 0 and 11 °C (Figure 15b,c). For the high $Z_H$ area, which is approximately 50 dBZ in the ML, it is noted that there is a negative $Z_{DR}$ area on the east side (higher), which is consistent with the characteristics of snow riming in previous studies [27,28]. The $U_{ar}$ shows an appearance similar to that of the X-band in Section 3.1, where a wide range of large values appears mostly just below the ML. This result indicates that $U_{ar}$ also has the ability to identify raindrop areas in S-band volume scan data.

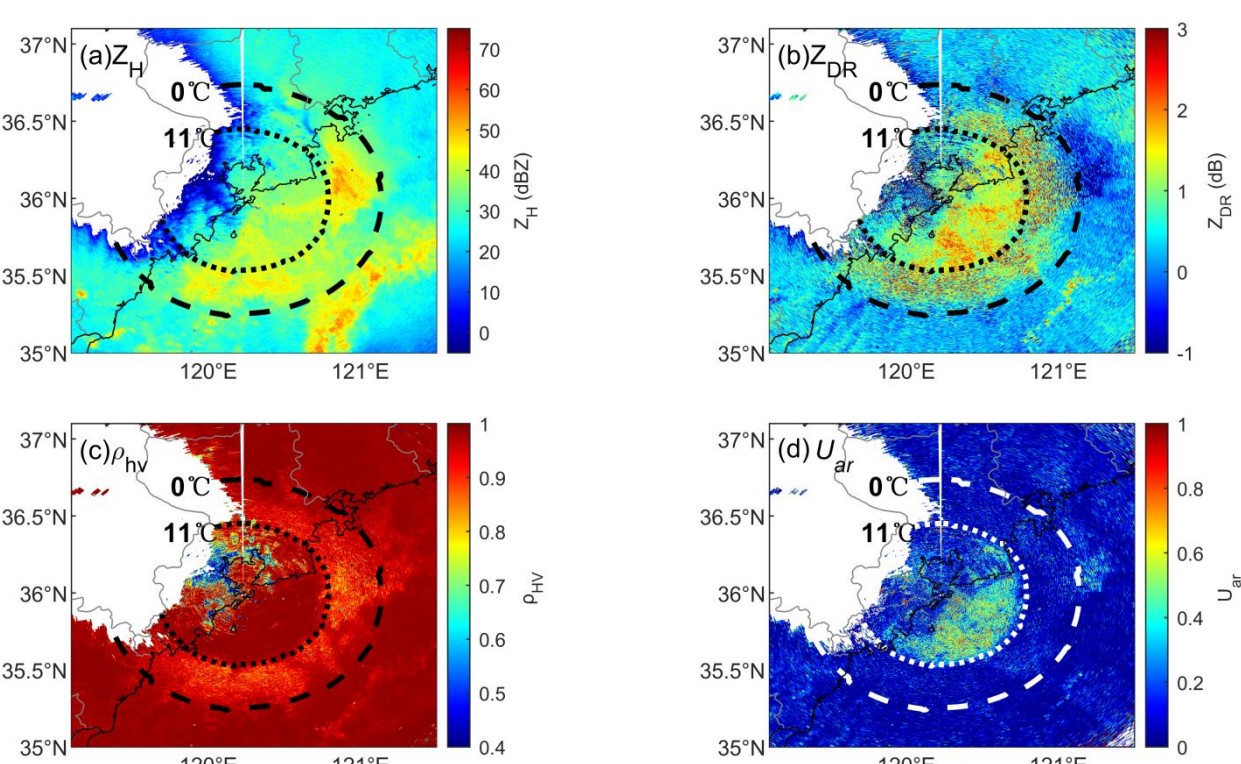

**Figure 15.** Sample PPI observations with BB from the S-band dual polarization weather radar. (The data time is 17 May 2020 23:51 LST. The elevation is 2.4°, which is the third level of a volume scan). (**a**) $Z_H$, (**b**) $Z_{DR}$, (**c**) $\rho_{hv}$, (**d**) $U_{ar}$.

To be further compared with the vertical structure in Section 3.1, a composite RHI is derived by interpolation, which covers both convective and stratiform areas (Figure 16). In the stratiform cloud area (distance from 0 to 60 km), the ML is visually estimated by the BB in $Z_H$, $Z_{DR}$ and $\rho_{hv}$, whose bottom is at the height of 11 °C, and the high $U_{ar}$ appears below the ML. The $Z_{DR}$ column at a distance of 100 km also corresponds to a high value

area of $U_{ar}$. An approximate U-shaped vertical structure of $U_{ar}$, whose two branches are at 70 and 100 km distances (Figure 16d), is more difficult to identify compared with that in X-band RHI data (Figure 12d). This may be due to the low elevation resolution of the volume scans. However, the general performances in the X-band and S-band are similar, regardless of the scanning mode.

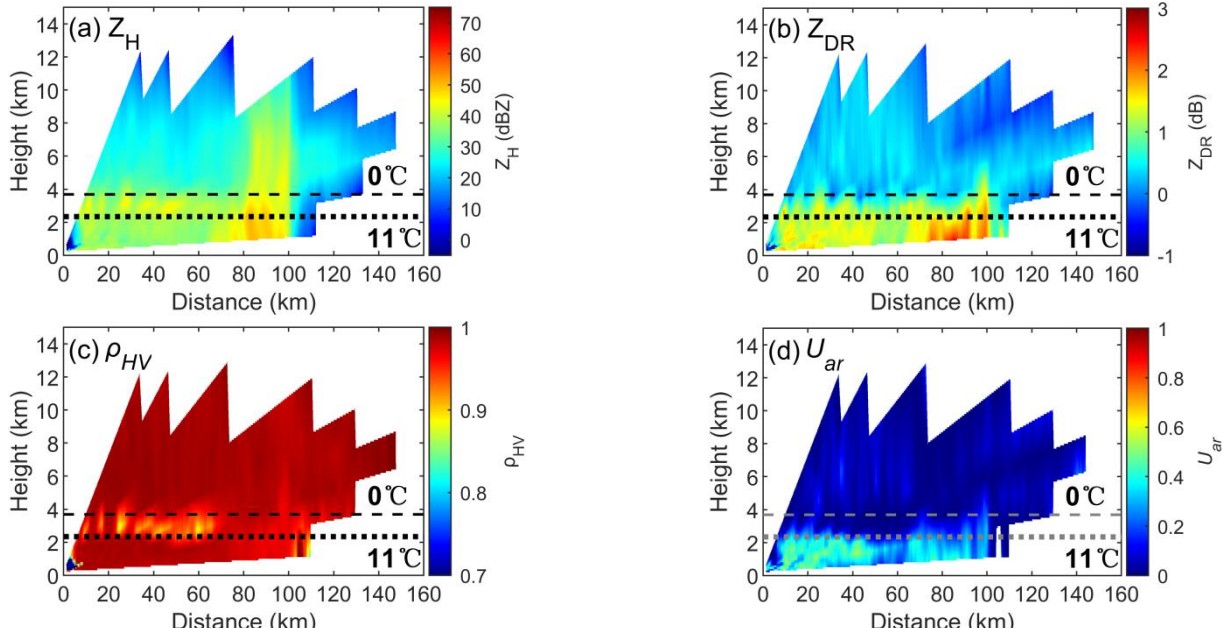

**Figure 16.** Composite RHI at 130° azimuth from the S-band dual polarization weather radar. (The data time is 17 May 2020 23:51 LST, the same as Figure 15. The radial data are smoothed by 10-point median filtering, and triple linear interpolation is used to derive this composite result). (**a**) $Z_H$, (**b**) $Z_{DR}$, (**c**) $\rho_{hv}$, (**d**) $U_{ar}$.

### 3.3. Identification Ratio of Raindrops in Stratiform Cloud Areas

One inevitable question is how accurate it is to use $U_{ar}$ to identify raindrops. However, the accurate phase state of particles in clouds is not easy to obtain, especially in convective areas, which is also the key and difficult point in the study of weather radar remote sensing. After all, there is not always a cloud-penetrating detection by aircraft to make a space–time continuous observation. Therefore, a relatively reliable method is selected. Aiming at stratiform areas with ML, the bottom height of the visual ML boundary is selected as the dividing line. Parts below the ML bottom are divided into rain layers, while other parts are divided into nonrain layers (mixed and ice phases), so that the identification results can be examined quantitatively. The data in Figure 10 (Section 3.1) are selected as Case 1, with the height of 9 °C as the dividing line; the data in Figure 16 (Section 3.2) are selected as Case 2, with the height of 11 °C as the dividing line. Weak echoes with $Z_H$ less than 20 dBZ are ignored in the statistics. The identification ratio of rain layers ($S_{rain}$), nonrain layers ($S_{nonrain}$) and overall ratio ($S_{total}$) are defined as follows:

$$S_{rain} = \frac{T_r}{N_r} \tag{26}$$

$$S_{nonrain} = \frac{T_i}{N_i} \tag{27}$$

$$S_{total} = \frac{T_r + T_i}{N_r + N_i} \tag{28}$$

where $N_r$ is data count in rain layers; $N_i$ is data count in nonrain layers; $T_r$ is data count of the correct identification of the rain layer, where a data point has a $U_{ar}$ larger than the

given threshold and is located in rain layers; $T_i$ is data count of correct identification of nonrain layer, where a data point has a $U_{ar}$ equal to or less than the given threshold and is located in nonrain layers.

The results (Table 3) show that $S_{rain}$ decreases as the $U_{ar}$ threshold increases, probably because small raindrops have a smaller $U_{ar}$ and cannot be identified, while the trend in $S_{nonrain}$ is the opposite. This results in a maximum value for $S_{total}$, where $U_{ar} = 0.3$ for Case 1 and $U_{ar} = 0.2$ for Case 2. In general, a more than 95% overall identification accuracy can be obtained when $U_{ar}$ is set to 0.2~0.3.

**Table 3.** Identification ratio of rain and nonrain layers using different thresholds of $U_{ar}$.

| Threshold of $U_{ar}$ | Case 1 | | | Case 2 | | |
|---|---|---|---|---|---|---|
| | $S_{total}$ | $S_{rain}$ | $S_{nonrain}$ | $S_{total}$ | $S_{rain}$ | $S_{nonrain}$ |
| 0.1 | 0.88 | 1.00 | 0.79 | 0.90 | 0.98 | 0.88 |
| 0.2 | 0.95 | 1.00 | 0.91 | 0.95 | 0.93 | 0.96 |
| 0.3 | 0.96 | 0.97 | 0.95 | 0.93 | 0.73 | 0.99 |
| 0.4 | 0.95 | 0.91 | 0.98 | 0.86 | 0.38 | 1.00 |
| 0.5 | 0.88 | 0.74 | 0.99 | 0.78 | 0.03 | 1.00 |

*3.4. Using $U_{ar}$ as a Mask to Compute Composite Reflectivity*

Composite reflectivity (CR) is a common data product of weather radar volume scan data and is a two-dimensional image derived from the $Z_H$ maximum at the same horizontal position in each PPI for different elevations. By using the CR, the horizontal spatial distribution of the strong reflectivity population can be quickly observed, and a preliminary judgment of precipitation can be made, where it can avoid missing information caused by a partial occlusion in a single PPI or the uncertainty of the height of the reflectivity core. Using the same case in Section 3.2, Figure 17a shows a CR, where there is a severe convective line over 60 dBZ and a large-scale stratiform cloud area at the rear (northwest side). However, due to the high value and uneven horizontal distribution of $Z_H$ in the ML, some spots in the ML may be misjudged as convective clouds, which would lead to misinterpretations of the precipitation situation. To avoid this, the CR must be recalculated after removing the effects of ML signals.

First, there is an example in which ML signals are not successfully removed. Suppose the lower bound temperature of the ML is 5 °C by assuming some experience that is not applicable to this case. After masking data with temperatures below 5 °C, a new CR is derived (Figure 17b), where the high $Z_H$ spots still exist in the stratiform area since the ML signal is not completely removed from the original PPI. This problem can certainly be solved if the lower bound of the ML in this example is accurately obtained at 11 °C. However, inaccuracies in the temperature profile and ML boundary detection can make a difference.

Next, $U_{ar}$ is taken as a mask template from $Z_H$ in each PPI. The $Z_H$ is masked where the $U_{ar}$ is less than a certain value, and then, the CR is calculated, which is equivalent to calculating the CR of raindrop areas. A loose threshold is applied first by taking $U_{ar} < 0.2$ as the mask template to form the new CR (Figure 17c). The intensity of the CR in the stratiform cloud region to the northwest of the strong convective line is obviously smaller and more uniform and is basically not more than 45 dBZ, which is consistent with the common features of stratiform rainfall. If a stricter threshold is used, such as by taking $U_{ar} < 0.4$ as the mask template (Figure 17d), the entire convective system becomes fragmented, and only part of the stratiform cloud region remains. The parts masked in Figure 17d may contain both smaller raindrops and particles, such as graupel and hail, that are likely to exist in the convective line. However, due to the need for other ground observation instruments to verify the identification of graupel and hail, such content is not discussed in this paper. In general, taking a loose $U_{ar}$ threshold as a mask (e.g., $U_{ar} < 0.2$) to calculate CR can preserve the horizontal spatial distribution characteristics of most precipitation systems while masking ML signals. Furthermore, this approach is more convenient and efficient

than traditional methods that require temperature input and multithreshold management to first detect the ML.

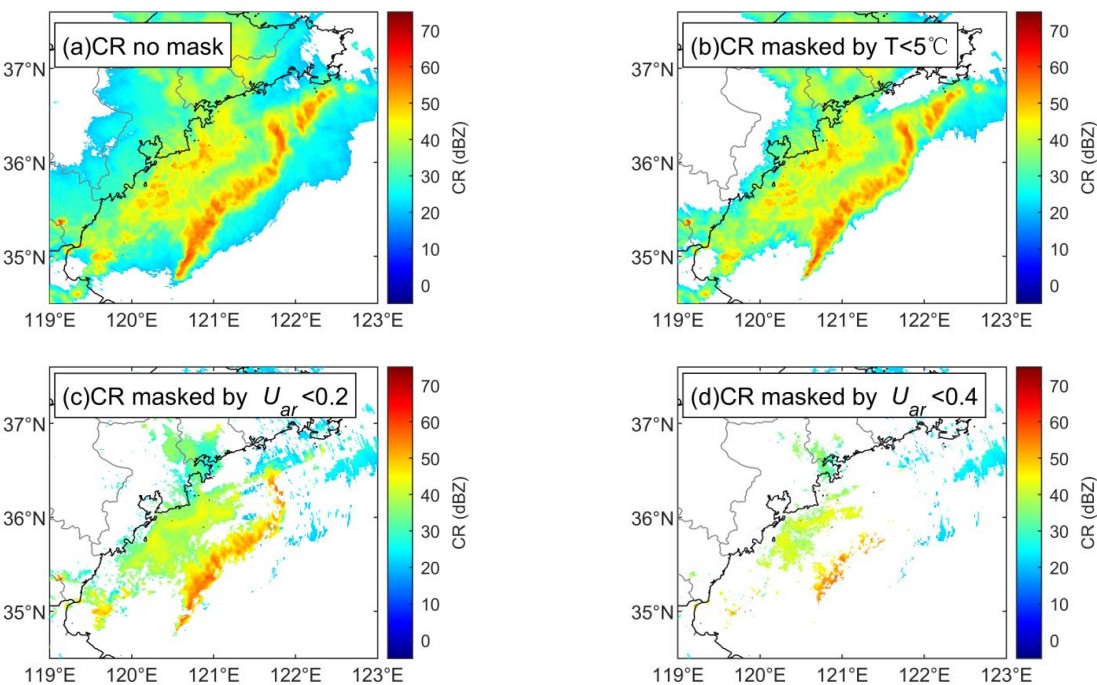

**Figure 17.** CR with different reflectivity masks. (The points are with $Z_H > 20$ dBZ. The time is the same as Figure 15. (**a**) CR no mask, (**b**) CR masked by T < 5 °C, (**c**) CR masked by $U_{ar} < 0.2$, (**d**) CR masked by $U_{ar} < 0.4$.

## 4. Discussion on Limitations of $U_{ar}$

There are also some limitations of using $U_{ar}$ that were found in the course of this study. Here are three points to consider.

(1) If the $Z_{DR}$ of the X-band does not undergo quality control (QC) and attenuation correction, there may be a large area of small $Z_{DR}$ anomalies in the lower troposphere away from the radar side due to attenuation (Figure 18a). In addition, in the area where the signal-to-noise ratio is theoretically low at the end of the radar beam, there may be an abnormally large or small $Z_{DR}$ value due to noise. These can result in a larger value and overestimate $U_{ar}$ (Figure 18b), which inevitably affects the results of raindrop area identification. One solution is to set $U_{ar}$ to 0 when $Z_{DR}$ is less than 0 dB. Thus, a rough location of the raindrop areas can be obtained without waiting for a time-consuming quality control and attenuation correction process when collecting data.

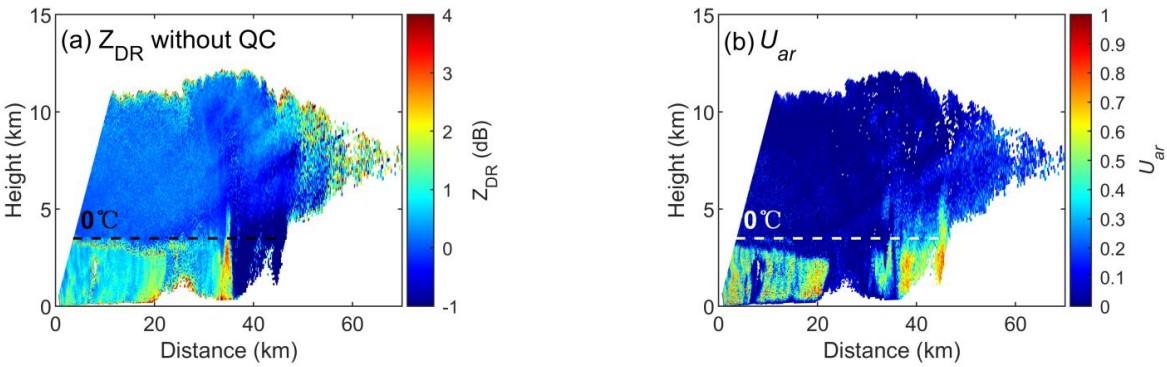

**Figure 18.** Sample of the impact on the $U_{ar}$ calculation without data quality control and attenuation correction for $Z_{DR}$ in the X-band. (**a**) $Z_{DR}$ without QC, (**b**) $U_{ar}$.

(2) If there is a systematic bias in the $Z_{DR}$ detected by the radar, it may cause $U_{ar}$ to be unavailable. A test in which a 0.3 dB systematic bias is artificially added to $Z_{DR}$ (Figure 19a) shows that $U_{ar}$ in the stratiform area no longer exhibits mutation features as shown in Figure 10, but rather displays a valley band feature (Figure 19b) similar to $\rho_{hv}$ in the ML and loses the capability to identify raindrop areas. This suggests that the premise of using $U_{ar}$ is that the systematic deviation of $Z_{DR}$ needs to be controlled within 0.3 dB. For these reasons, attention should be given to the calibration of dual polarization radars, especially for mobile X-band radars, when conducting field observation experiments.

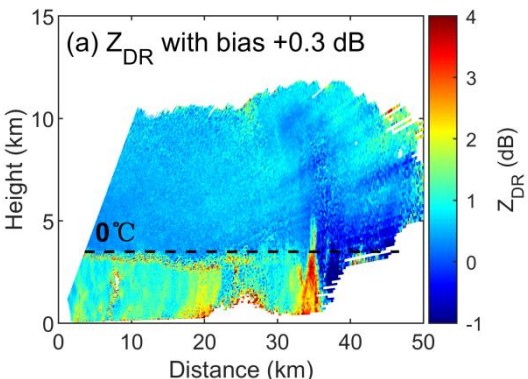
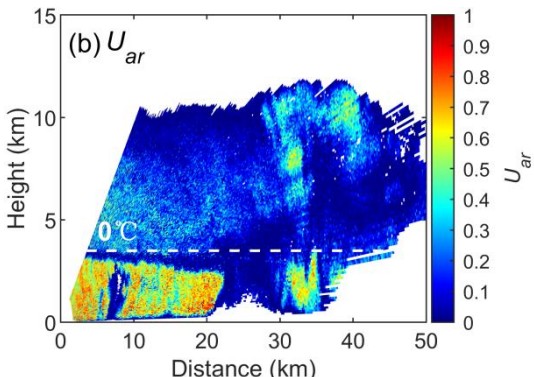

**Figure 19.** Sample of the impact on the $U_{ar}$ calculation when there is a +0.3 dB systematic deviation in $Z_{DR}$ at the X-band. (**a**) $Z_{DR}$ with bias +0.3 dB, (**b**) $U_{ar}$.

(3) The premise of using $U_{ar}$ is to eliminate or at least reduce the influence of the phase state and only to retrieve and utilize the axial ratio distribution characteristics of the particle group. However, there must be some extreme cases where the ice or snow particles exhibit a pronounced horizontal orientation under the action of the dominant wind. This could also lead to large values of $U_{ar}$, which could be confused with the raindrop area. However, there is no good example to illustrate this expected extreme situation, and this work needs to be carried out in depth in the future. In addition, the cases presented in this paper involve relatively low elevations, and observations at high elevations will result in a smaller $Z_{DR}$ in the raindrop area and would need to be corrected. The impact of these factors on $U_{ar}$ also needs to be explored in more cases.

## 5. Conclusions and Summary

A uniformity index for hydrometeor axis ratios ($U_{ar}$) derived from dual polarization weather radar data is proposed in this paper. Backscattering numerical simulations are used to find available relationships to derive $U_{ar}$ and show its theoretical features for the identification of raindrops. Then, observation data from X-band and S-band radar are used to show and examine the performance of $U_{ar}$ under real conditions and carry out initial applications. The main conclusions are as follows.

(1) $U_{ar}$ is close to 0 for ice particles with varying shapes and orientations and is close to 1 for raindrops theoretically, which gives $U_{ar}$ the ability to identify raindrop areas.

(2) In the real observations, $U_{ar}$ is basically consistent with its theoretical feature above. A more than 95% overall identification ratio can be obtained in stratiform cloud areas when the threshold of $U_{ar}$ is set to 0.2~0.3. Thus, the raindrop area can be more easily identified instead of identifying ML first by inputting a temperature profile and setting multiple thresholds.

(3) In the demonstration using X-band radar RHI data, high $U_{ar}$ in the convective areas presents a U-shaped vertical structure, which indicates the process of ice particles melting into raindrops and then being transported upward by strong updraft and provides evidence for the formation mechanism of the $Z_{DR}$ column.

(4) In the CR calculation demonstration using S-band radar, the impact of the ML signal on CR can be eliminated by setting $U_{ar} < 0.2$ as a mask template to avoid misjudging stratiform clouds at the rear of the convective line as convective clouds.

The application of $U_{ar}$ still requires more in-depth research in the future. Due to the spatiotemporal limitations of RHI and volume scanning, the change process of the $Z_{DR}$ column may not be fully captured. Thus, more studies are needed to better summarize the evolution of $U_{ar}$ and other variables. Additionally, it is necessary to further evaluate and expand the application of $U_{ar}$ methods in hydrometeor classification and in the quantitative retrieval of microphysical features.

**Author Contributions:** Conceptualization, Y.S.; methodology, Y.S.; software, L.F. and W.S.; validation, Y.S. and L.F.; formal analysis, Y.S.; investigation, Y.S.; resources, H.X., H.C. and H.Y. (Han Yao); data curation, H.Y. (Han Yao); writing—original draft preparation, Y.S.; writing—review and editing, H.X., H.C. and H.Y. (Huiling Yang); visualization, W.S.; supervision, H.X.; project administration, H.X.; funding acquisition, Y.S. and H.X. All authors have read and agreed to the published version of the manuscript.

**Funding:** This research was supported in part by the National Key Research and Development Plan of China (grant no. 2019YFC1510304), the National Natural Science Foundation of China (grant no. 42105127), and the Special Research Assistant Project of Chinese Academy of Sciences.

**Data Availability Statement:** Not applicable.

**Conflicts of Interest:** The authors declare no conflict of interest.

## Appendix A

In Section 2.2 (Figure 5), R, MAE and MRE are used to evaluate the difference between $\rho_{hv}^{(Ideal)}$ and $\rho_{hv}^{(Aprrox)}$, which are defined as follows:

$$R = \frac{Cov(X, Y)}{\sqrt{Var(X) \cdot Var(Y)}} \tag{A1}$$

$$MAE = \frac{\sum_{i=1}^{n} |Y_i - X_i|}{n} \tag{A2}$$

$$MRE = 100\% \times \sum_{i=1}^{n} \left| \frac{|Y_i - X_i|}{Y_i} \right| \tag{A3}$$

where X is $\rho_{hv}^{(Aprrox)}$, Y is $\rho_{hv}^{(Aprrox)}$, $Cov(...)$ denotes the covariance and $Var(...)$ indicates the variance.

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
