# Peer review of "A Uniformity Index for Precipitation Particle Axis Ratios Derived from Radar Polarimetric Parameters for the Identification and Analysis of Raindrop Areas"

_remotesensing, doi:10.3390/rs15020534_

Round 1
Reviewer 1 Report
This is an innovative work.
Some comments:
1) Some radar parameters should be introduced, such as transmitting power, beamwidth, operation mode, etc.
2) Through radar data processing, the experimental result of the identification ratio of raindrop by the new method should be given
Reviewer 2 Report
In this work the authors have identified, by means of some simulations, a quantity observable by means of radar measurements able to discriminate the liquid part of the precipitation.
The authors tested the proposed method on two case studies.
The work is done accurately and the exposure is very good.
It is recommended that the authors test this method on other case studies in the future (two are few).
Minor remarks
Equation 8.
Who is a0? Maybe it's a2 from the formula Y = a1X + a2?

Reviewer 3 Report
This manuscript (Title: A Uniformity Index for Precipitation Particle Axis Ratios Derived from Radar Polarimetric Parameters for the Identification and Analysis of Raindrop Areas; ID
remotesensing-2125710) investigates a uniformity index used to identify and analyze raindrop areas. The problem statement is clear, and the research content is substantial. According to the existing presentation, the following aspects need to be improved.
1. Is there any special research background for this study? Why we should focus on discussing the uniformity index of the axial ratio of precipitation particles?
2. The literature involved in the presentation in the introduction is few. It is suggested to give more explanation to relevant historical research, including the analysis of reflectivity and different cloud and rain layers.
3. For formula (1), the symbol "dot multiplication" is not used between numbers and letters or operators. Please check the expression of other formulas.
4. Whether the simulation data is compared with the detection data of weather radar?
5. In line 150, the equivalent spherical diameter (D) of the particle should be italicized.
6. Conclusion writing is lengthy, and relevant key results need to be refined to draw 3 to 4 general arguments.
7. The numbering format of reference 48 at the end of the article is different.
Round 2
Reviewer 3 Report
For this manuscript (ID :remotesensing-2125710),the authors have made a lot of efforts to improve all aspects of the content. The statements in the introduction are more sufficient and extensive, and the statements of variables are more standardized and reasonable. As a result, the discussion has also been improved.
1. In line 456, Z in ZDR should be italicized. (Also in line 458)
2. Line 348 and 351, noting the format of ZH and ZDR.